TROPICAL DISEASES

# Heatwaves cause fluctuations in *w*Mel *Wolbachia* densities and frequencies in *Aedes aegypti*

**Perran A. Ross**[1]*, **Jason K. Axford**[1], **Qiong Yang**[1], **Kyran M. Staunton**[2,3], **Scott A. Ritchie**[2,3], **Kelly M. Richardson**[1], **Ary A. Hoffmann**[1]

**1** Pest and Environmental Adaptation Research Group, Bio21 Institute and the School of BioSciences, The University of Melbourne, Parkville, Victoria, Australia, **2** College of Public Health, Medical and Veterinary Sciences, James Cook University, Smithfield, Queensland, Australia, **3** Australian Institute of Tropical Health and Medicine, James Cook University, Smithfield, Queensland, Australia

☯ These authors contributed equally to this work.
* perran.ross@unimelb.edu.au

**Data Availability Statement:** All relevant data are within the manuscript and its Supporting Information files.

## Abstract

*Aedes aegypti* mosquitoes infected with the *w*Mel strain of *Wolbachia* are being released into natural mosquito populations in the tropics as a way of reducing dengue transmission. High temperatures adversely affect *w*Mel, reducing *Wolbachia* density and cytoplasmic incompatibility in some larval habitats that experience large temperature fluctuations. We monitored the impact of a 43.6°C heatwave on the *w*Mel infection in a natural population in Cairns, Australia, where *w*Mel was first released in 2011 and has persisted at a high frequency. *Wolbachia* infection frequencies in the month following the heatwave were reduced to 83% in larvae sampled directly from field habitats and 88% in eggs collected from ovitraps, but recovered to be near 100% four months later. Effects of the heatwave on *w*Mel appeared to be stage-specific and delayed, with reduced frequencies and densities in field-collected larvae and adults reared from ovitraps but higher frequencies in field-collected adults. Laboratory experiments showed that the effects of heatwaves on cytoplasmic incompatibility and density are life stage-specific, with first instar larvae being the most vulnerable to temperature effects. Our results indicate that heatwaves in *w*Mel-infected populations will have only temporary effects on *Wolbachia* frequencies and density once the infection has established in the population. Our results are relevant to ongoing releases of *w*Mel-infected *Ae. aegypti* in several tropical countries.

## Author summary

Mosquitoes infected with *Wolbachia* bacteria are being released in the tropics to replace natural mosquito populations and suppress dengue transmission. *Aedes aegypti* mosquitoes with the *w*Mel strain of *Wolbachia* were first released in Cairns, Australia in 2011 and releases were then expanded to the entire city and surrounding suburbs. Today, *w*Mel is at a high frequency within the *Ae. aegypti* population and local dengue transmission in Cairns has declined to nearly zero. *Wolbachia* infections are vulnerable to high

**Funding:** AAH was supported by the National Health and Medical Research Council (1132412, 1118640, www.nhmrc.gov.au). The funders had no role in study design, data collection and analysis, decision to publish, or preparation of the manuscript.

**Competing interests:** The authors have declared that no competing interests exist.

temperatures and the ability of *w*Mel to persist in populations and block dengue may be constrained by climate. Cairns experienced a record heatwave of 43.6˚C in November 2018 and we wanted to see whether this affected the *w*Mel-infected *Ae. aegypti* population. Our results show that the frequency and density of *w*Mel declined after the heatwave, with effects depending on the mosquito life stage tested. When we monitored the population again in April 2019, *w*Mel had returned to a high frequency. We suggest that heatwaves of the magnitude experienced in Cairns will not have long-term impacts on the *w*Mel infection but may affect invasion during releases or interfere with dengue blockage. Heatwaves may affect interventions with *w*Mel-infected *Ae. aegypti* that are being deployed in several countries. Effects may depend on the proportion of larval habitats that are protected from extreme temperature fluctuations.

## Introduction

*Aedes aegypti* mosquitoes with novel *Wolbachia* infections are increasingly being deployed for disease control [1]. These programs rely on cytoplasmic incompatibility induced by *Wolbachia*, where uninfected females produce few or no viable offspring when mated to *Wolbachia*-infected males. When only male mosquitoes are released this can lead to population suppression, while the release of males and females can drive population replacement by *Wolbachia*-infected mosquitoes [2, 3]. Many *Wolbachia* infections introduced into *Ae. aegypti* also reduce the capacity for mosquitoes to transmit arboviruses [4–7]. Populations replaced with *Wolbachia*-infected mosquitoes can therefore lead to reduced arbovirus transmission without mosquito population suppression [8].

In *Ae. aegypti*, releases of two *Wolbachia* strains are currently underway. Mosquitoes with the *w*AlbB infection have been released for population replacement in Malaysia [9] and for population suppression in Australia [10], Singapore (https://www.nea.gov.sg/corporate-functions/resources/research/wolbachia-aedes-mosquito-suppression-strategy/project-wolbachia-singapore) and the USA (https://verily.com/projects/interventions/debug/). Releases of the *w*Mel strain for population replacement have been undertaken in at least 10 countries including Australia and Brazil [11, 12]. Preliminary reports indicate replacement releases have led to reduced dengue transmission [8, 9, 13], while population suppression releases in *Ae. albopictus* led to reduced mosquito populations and biting rates [14]. The success of replacement programs will depend on the long-term stability of *Wolbachia* infections in populations [1, 15]. For robust dengue suppression, *Wolbachia* infections should remain at high frequencies in populations and at a high density within individual mosquitoes. It is therefore vital to understand the factors that can affect the stability of *Wolbachia* infections in populations.

*Wolbachia* infections are vulnerable to temperature extremes that are often within the range of temperatures that their hosts can tolerate [16]. *Wolbachia* strains in *Ae. aegypti* differ in their vulnerability to heat stress, with the *w*AlbB infection being more robust than *w*Mel under laboratory conditions [17]. High temperatures reduce *w*Mel density, with effects depending on the life stage and the duration of exposure [18]. Effects on density are apparent at maximum daily temperatures as low as 32˚C over long exposure periods [19]. While effects can persist across generations [20], *Wolbachia* infections can also recover under cooler conditions [18]. Reduced *Wolbachia* density results in weaker cytoplasmic incompatibility, maternal transmission failure [7, 17] and potentially reduced virus blockage [21]. In the field, *w*Mel-infected *Ae. aegypti* exposed to high temperatures during development have reduced density

and weaker cytoplasmic incompatibility [22]. Critical temperatures resulting from heatwaves or from breeding in certain locations could therefore reduce *Wolbachia* infection frequencies and virus blocking potential in the population, affecting the success of *w*Mel release programs.

Although the *w*Mel infection is vulnerable to high temperatures, this strain has been released successfully in several tropical locations where temperatures may have a negative effect on *Wolbachia*. Releases in Brazil led to the successful establishment of *w*Mel in *Ae. aegypti* despite high temperatures during releases [12], but temperature may in part explain the maternal transmission failure [12, 23] and fluctuating infection frequencies [8, 12, 13] observed in some release locations. To date, no studies have directly tested for relationships between temperature and *Wolbachia* infections in natural mosquito populations. Since the success of *Wolbachia* release programs depends on the stability of *Wolbachia* strains in nature, monitoring populations for temperature effects will be important, particularly as climate change is leading to increasing temperature extremes around the world [24].

In Cairns, Australia, where *w*Mel-infected *Ae. aegypti* were first released into the field in 2011 [11], average monthly maximum temperatures typically reach 31.5˚C in mid-summer (http://www.bom.gov.au/climate/averages/tables/cw_031011.shtml). In November 2018, Cairns experienced a heatwave with three consecutive days over 40˚C which included its hottest day on record. With the *w*Mel infection established at a high frequency in natural populations in Cairns [8], we were interested in monitoring the stability of *w*Mel in response to this extreme weather event. Through field sampling, we tested to see if *Wolbachia* infection frequencies and densities were reduced following the heatwave and if recovery occurred in later months. We then performed laboratory experiments to determine which life stages were most vulnerable to simulated heatwaves, and to test for associations between *Wolbachia* density and cytoplasmic incompatibility. Our results can inform the choice of *Wolbachia* strains and the timing of releases in different tropical environments.

## Methods

### Ethics statement

Blood feeding of female mosquitoes on human volunteers for this research was approved by the University of Melbourne Human Ethics Committee (approval 0723847). All adult subjects provided informed written consent (no children were involved)."

### Field sampling

We placed 150 ovitraps around central suburban Cairns (with traps in Bungalow, Portsmith, Parramatta Park, Cairns North and Edge Hill) from February to March 2018 to determine a baseline *Wolbachia* infection frequency. Verbal permission was obtained from each household before placing a single ovitrap in a covered location on the premises. Ovitraps consisted of 1 L black plastic buckets half-filled with water and lined with a strip of red felt (50 mm x 100 mm). Lucerne pellets were placed in ovitraps to encourage mosquito oviposition [25]. One week after deployment, ovitraps were collected and felt strips were partially dried and stored at a high humidity.

Eggs were hatched in the laboratory approximately one week after collection by submerging felt strips in plastic trays with 500 mL of water. Hatching larvae were provided with TetraMin tropical fish food tablets (Tetra, Melle, Germany) *ad libitum*. Three days after hatching, larvae were identified to the species level using an identification key [26] and other species were discarded. The larval density was controlled to a maximum of 100 larvae per tray to ensure synchronous development and reduce the potential effects of larval crowding on *Wolbachia* density. All *Ae. aegypti* larvae hatching from each felt strip were reared to the 4th larval instar

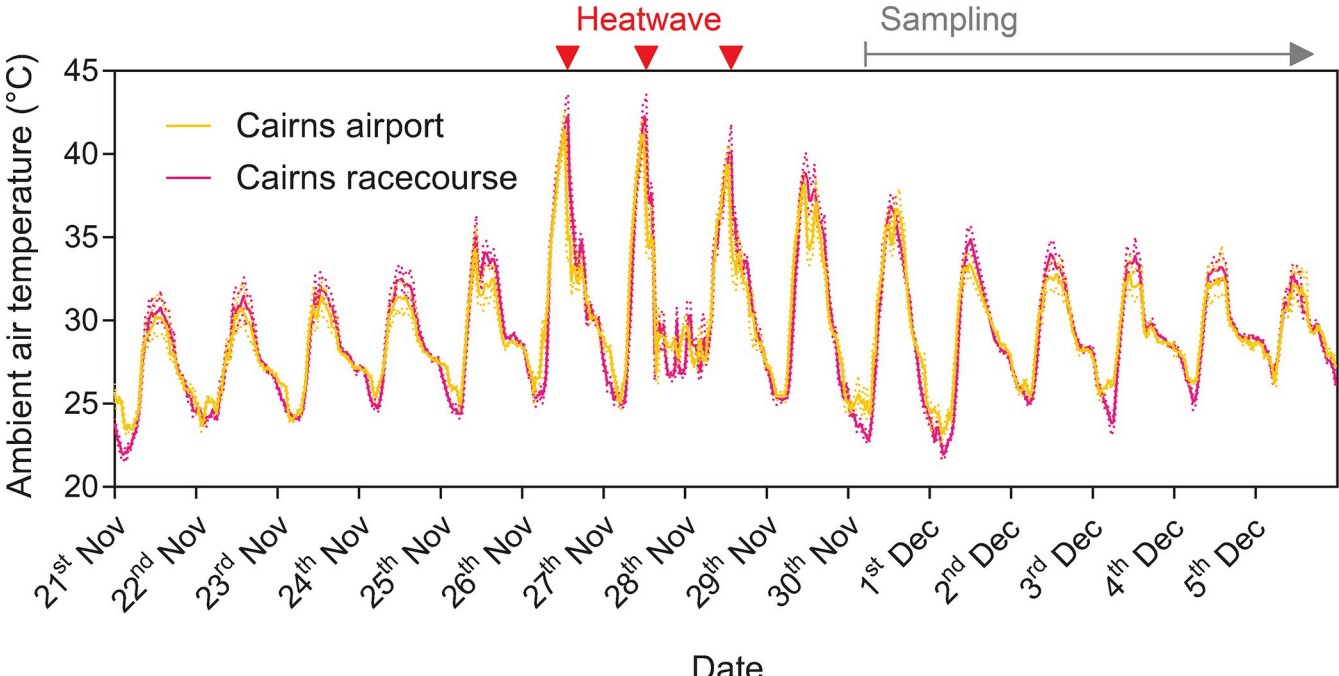

**Fig 1. Ambient air temperatures recorded at two stations in Cairns during the 2018 heatwave.** Data were recorded at 1-minute intervals. 30-minute average temperatures are shown by solid lines, with dotted lines representing the maximum and minimum temperatures recorded during each 30-minute period. Red arrows indicate days where temperatures exceeded 40˚C, while the gray arrow indicates when sampling commenced. Temperature data were provided by the Bureau of Meteorology (www.bom.gov.au).

stage and stored in 100% ethanol. Samples were then screened for the presence and density of *w*Mel (see "*Wolbachia* detection and density quantification" below), with up to two larvae tested per ovitrap.

Field sampling of *Ae. aegypti* was conducted in December 2018 following a heatwave in Cairns, Queensland, Australia in late November. Ambient air temperatures were obtained from the Bureau of Meteorology (www.bom.gov.au) for the Cairns racecourse and Cairns airport stations which are both within 3 km of the *w*Mel release zone and should have similar ambient temperatures. Cairns experienced its hottest day on record on the 26[th] of November 2018, with maximum ambient air temperatures of 43.6˚C (Cairns racecourse) and 42.6˚C (Cairns airport) and three consecutive days over 40˚C (Fig 1).

Sampling of all life stages took place within the central suburbs of Cairns where the *w*Mel infection had established. The *w*Mel releases in this area began in 2013 and continued until 2017 [8], with complete coverage achieved by early 2018 (see results). Adults, larvae and pupae were collected beginning on the 30[th] of November 2018 and throughout the entire month of December 2018. We collected adult *Ae. aegypti* using Biogents Sentinel (BGS) traps, sweep nets and prototype acoustic traps which targeted males using sound lures set to 500 Hz and 60 dB at the trap entrance, consistent with similar traps [27, 28]. Live larvae and pupae were collected from larval habitats (e.g. tires, dog bowls) in the field. All specimens collected from the field were stored in 100% ethanol at 4˚C for *Wolbachia* screening (see "*Wolbachia* detection and density quantification" below). *Wolbachia* density can vary throughout development (see results); we therefore included only 4[th] instar larvae and adults in our analyses of density since sample sizes of other stages were low. However, all individuals were included in our assessment of *Wolbachia* infection frequencies.

To assess the potential effects of the heatwave on *Wolbachia* densities and frequencies in the subsequent generation, we conducted ovitrapping immediately following the heatwave in late November 2018 and again in April 2019 when temperatures were milder, with an average daily maximum temperature of 29.4°C. Forty ovitraps were placed throughout central Cairns on November 30th in the same suburbs as earlier in the year. Ovitraps were collected each week and replaced for a total of three consecutive weeks of sampling. Ovitrapping was conducted again in 2019 when 36 traps were deployed from the 3rd to the 5th of April in similar locations to those in the previous year. Methods were identical to the sampling conducted in February-March 2018 except that larvae were reared to adulthood and stored within 24 hr of emergence, with up to 6 males and 6 females from each ovitrap screened for *Wolbachia* (see "*Wolbachia* detection and density quantification" below).

## Sentinel containers

We monitored containers placed within the *w*Mel release zone that were colonized naturally by *Ae. aegypti*. In January, black plastic buckets were placed in either full shade or in an area exposed to sunlight next to a garden bed. Buckets were filled with 8 L of water and provided with a few lucerne pellets to encourage mosquito oviposition and to hasten larval development. Buckets were checked approximately twice per week until *Ae. aegypti* larvae were detected. Data loggers (Thermochron; 1-Wire, iButton.com, Dallas Semiconductors, Sunnyvale, CA, USA) placed in sealed zip-lock bags were submerged in each container when confirmed as a positive *Ae. aegypti* habitat, with temperature recorded at 30-minute intervals. Pupae were sampled from the container in sunlight on the 23rd of January and the 1st, 4th and 5th of February and from the container in full shade on the 5th of February only. Pupae were returned to the laboratory and adults were stored in ethanol within 24 hr of emergence for *Wolbachia* density measurements. Between 5 and 12 adults of each sex were tested from each collection date.

## Mosquito strains and colony maintenance

We used uninfected and *w*Mel-infected *Ae. aegypti* in our laboratory experiments. An uninfected colony was established in the laboratory using $F_2$ eggs sourced from central Cairns, Australia. Females infected with *w*Mel were sourced from Gordonvale, Australia and were outcrossed for three generations to $F_4$ uninfected males prior to the start of the laboratory experiments. Colonies were maintained in an insectary at 26 ± 1°C with a 12 hr photoperiod and 1 hr dawn and dusk phases according to methods described previously [29, 30].

We reared *w*Mel-infected *Ae. aegypti* in the laboratory for comparisons of *Wolbachia* density with field-collected specimens. For these comparisons, *Ae. aegypti* were collected from central Cairns in 2018 and reared in the laboratory for at least three generations before testing.

## Simulated heatwave exposure across life stages

*w*Mel and uninfected eggs were collected from laboratory colonies on strips of Norton Master Painters P80 sandpaper (3.8 × 18 cm; Saint-Gobain Abrasives Pty. Ltd., Thomastown, Victoria, Australia). Eggs were completely dried by the third day post-oviposition to prevent hatching. Before commencing the experiment, sandpaper strips were sealed in multiple zip-lock bags to prevent water from reaching the eggs. Life stages were either unstressed (held in a temperature-controlled room at 26°C) or exposed to heatwaves in a water bath (Grant Instruments R2 refrigerated circulator with GP200 thermostat), where temperatures cycled daily between 26 and 39°C (Fig 2A). *w*Mel-infected populations were exposed to 3 d heatwaves that approximately covered the following life stages: eggs, early instar larvae, late instar larvae, and pupae (Fig 2B). Eggs were hatched on Day 4 of the experiment, immediately after the first 3 d

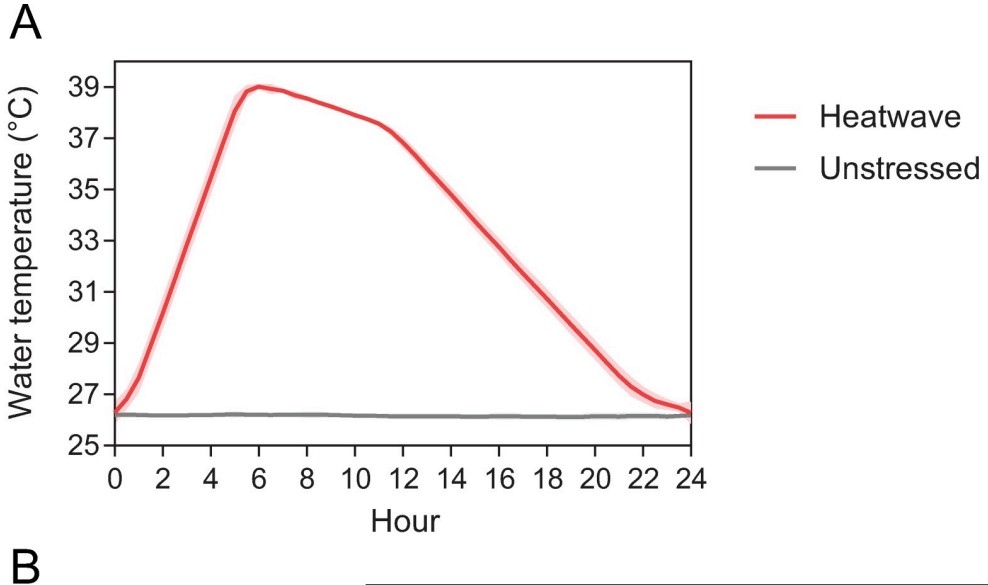

**Fig 2. Experimental design and temperature cycles for laboratory heatwave experiments.** (A) Temperature cycles experienced during heatwave (red) and unstressed (gray) conditions. Lines and shaded areas represent temperatures measured at 30-minute intervals and 95% confidence intervals, averaged across all days of the experiment. (B) Experimental design showing the timing and duration of exposure of *w*Mel-infected and uninfected populations to heatwave (red) or unstressed (gray) conditions at each life stage. "X" symbols indicate the treatments and time points where individuals were stored for *Wolbachia* density measurements.

heatwave treatment, by removing sandpaper strips from zip-lock bags and submerging them in RO (reverse osmosis) water with a few grains of yeast. Larvae were reared through to adulthood in trays with 500 mL water by providing TetraMin *ad libitum*. *w*Mel-infected controls that were unstressed, or stressed across all four stages were also reared. Uninfected populations were either unstressed or exposed to heatwaves across all stages for cytoplasmic incompatibility crosses (see below). Subsets of individuals from the unstressed group and groups exposed to single heatwaves or continuous heatwaves were stored immediately after each 3 d heatwave treatment had elapsed. Thirty individuals (larvae, pupae) were stored and tested for each time point, except for eggs where 20–23 individuals were tested.

## Cytoplasmic incompatibility

We performed crosses to (1) test the ability of *w*Mel-infected males that were exposed to heatwaves at different life stages to induce cytoplasmic incompatibility and (2) test the ability of *w*Mel-infected females that were exposed to heatwaves to restore compatibility. To test for cytoplasmic incompatibility induction, male adults emerging from each heatwave treatment were crossed with uninfected females. To test for compatibility restoration, females emerging from each heatwave treatment were crossed to unstressed, *w*Mel-infected males. Control crosses were also established to confirm that effects on egg hatch proportions were due to loss of *Wolbachia* infection and not due to direct effects of heatwaves on fertility [31]. Adults were up to 7 days old when crosses were established since development times varied between heatwave treatments. For each cross, we established 4 replicate cages (1.5 L) with 10 females and 10 males each. Cages were provided with 10% sucrose and water. Adults (starved for 24 hr) were blood-fed at approximately 11 days post-emergence by a single human volunteer (one of the authors of this study). Three days after blood feeding, each cage was provided with a cup filled with larval rearing water and lined with filter paper (Whatman 90 mm qualitative circles, GE Healthcare Australia Pty. Ltd., Parramatta, New South Wales, Australia). Eggs laid on filter papers were collected 4 days after blood feeding. Six days after collection, eggs were hatched by submerging filter papers in RO water with a few grains of yeast and 150 mg of TetraMin. Eggs on filter papers were photographed and counted using the multi-point tool in ImageJ Version 1.51j8 [32]. Larvae were counted 2–7 days after hatching, and hatch proportions were determined by dividing the number of larvae hatched by the number of eggs from each cage. Male and females used in crosses were stored in 100% ethanol 2 days after egg collections for *Wolbachia* density analysis.

## *Wolbachia* detection and density quantification

*Aedes aegypti* from the field sampling and laboratory experiments were tested for the presence and density of *w*Mel according to Lee et al. [33] via the Realtime LightCycler 480. Genomic DNA was extracted using 250 μL of 5% Chelex (Bio-Rad Laboratories, Hercules, CA) and 3 μl of Proteinase K (20 mg/mL) solution (Roche Diagnostics Australia Pty. Ltd., Castle Hill New South Wales, Australia). Tubes were incubated for 30 minutes at 65˚C then for 10 minutes at 90˚C. Three primer sets were used to amplify markers to confirm *Aedes* genus, *Ae. aegypti* species and the presence or absence of the *w*Mel infection [33]. Differences in crossing point (Cp) values between the *Ae. aegypti*-specific and *w*Mel markers were averaged from three consistent replicate runs and transformed by $2^n$ to produce relative *Wolbachia* density estimates. Samples that were inconclusive or negative for the *Ae. aegypti*-specific marker were excluded from the analyses since these were likely to be other *Aedes* species.

## Statistical analysis

All data were analyzed using SPSS Statistics version 24.0 for Windows (SPSS Inc, Chicago, IL). Data sets were tested for normality with Shapiro-Wilk tests. *Wolbachia* density data from field-collected larvae and adults, and egg hatch proportion data could not be transformed for normality and were analysed with Mann-Whitney U tests. General linear models (GLMs) were used to analyze *Wolbachia* density data from the laboratory experiments, the sentinel container experiment and from adults reared from ovitraps, where traps were nested within the collection date. To test for differences in *Wolbachia* density between the sexes in adults reared from ovitraps, we ran GLMs considering the effect of sex on both *Ae. aegypti* and *w*Mel Cp values. A single GLM was used to test for immediate effects of simulated heatwaves across life stages and separate GLMs were used to test for effects in male and female adults.

Differences in density between treatments, containers and collection dates were compared with Tukey's post-hoc tests with Bonferroni correction. Two-proportions z-tests were used to compare *Wolbachia* infection frequencies between collection periods from the field sampling. Spearman's rank-order correlations were used to test the relationships between *Wolbachia* density and egg hatch proportions in the laboratory experiments.

## Results

### Loss and recovery of *w*Mel frequencies and densities following a heatwave

The *w*Mel infection was at a high frequency in early 2018, with 98.48% of individuals infected (Table 1). We sampled larvae, pupae and adults directly from the field with sweep nets, BGS traps, acoustic traps and collections from larval habitats following the heatwave to test for any immediate effects on *Wolbachia* infection frequencies and densities in these life stages. Field-collected larvae and pupae had an infection frequency of 83.33% in December 2018 which was significantly lower than in early 2018 based on ovitrap data (two-proportions z-test: $\chi^2 = 24.278$, df = 1, P < 0.001). In contrast, adults collected in December 2018 had an infection frequency of 96.12%, similar to early 2018 ($\chi^2 = 0.793$, df = 1, P = 0.373, Table 1).

Adults collected in December 2018 had heterogeneous densities that were significantly lower than laboratory-reared adults (Mann-Whitney U: z = 3.762, P < 0.001, Fig 3A). Fourth instar larvae collected during the same period also exhibited variable densities that on average were similar to laboratory-reared larvae (z = 1.248, P = 0.211, Fig 3B). Adults (z = 5.626, P < 0.001) and 4th instar larvae (z = 3.205, P = 0.001) sampled in early 2019 had reduced densities compared to laboratory reared counterparts. Overall, these results indicate that that the heatwave has led to a temporary decrease in *w*Mel infection frequencies and densities, though only a subset of the population was affected.

We conducted ovitrapping to monitor the cross-generational impact of the November 2018 heatwave on *Wolbachia* densities and frequencies within the *w*Mel release zone in Cairns. On average, 70.8% of ovitraps were positive for *Ae. aegypti* with a median of 14 larvae per ovitrap (range: 1–220). In December 2018, the *Wolbachia* infection frequency declined to 87.62% early in the month and then increased over the next two weeks of sampling. Infection frequencies during December were significantly lower than in early 2018 for all three weeks of sampling (two-proportions z-test: all P < 0.035). In April 2019, infection rates had returned to near 100%, except for a single ovitrap where all individuals tested were uninfected. Given that this ovitrap was within the *w*Mel release zone and that *w*Mel has established even in non-release areas in Cairns [8], these uninfected individuals may have resulted from incomplete

**Table 1. *Wolbachia* infection frequencies in *Aedes aegypti* within the *w*Mel release zone, monitored through ovitrapping or direct sampling of larvae, pupae and adults.** The heatwave event occurred during the 26-29th of November 2018.

| Life stage | Collection period | Proportion *Wolbachia*-infected (number tested) | Binomial confidence interval (lower 95%, upper 95%) |
|---|---|---|---|
| Eggs | February-March 2018 | 0.985 (n = 198) | 0.956, 0.997 |
|  | December 7th 2018 | 0.876 (n = 210) | 0.824, 0.918 |
|  | December 14th 2018 | 0.913 (n = 254) | 0.872, 0.945 |
|  | December 21th 2018 | 0.933 (n = 238) | 0.893, 0.961 |
|  | April 2019 | 0.978 (n = 226) | 0.949, 0.993 |
| Larvae and pupae | December 2018 | 0.833 (n = 144) | 0.762, 0.890 |
|  | February 2019 | 1.000 (n = 53) | 0.933, 1.000 |
| Adults | December 2018 | 0.961 (n = 103) | 0.904, 0.989 |
|  | January-February 2019 | 1.000 (n = 42) | 0.916, 1.000 |

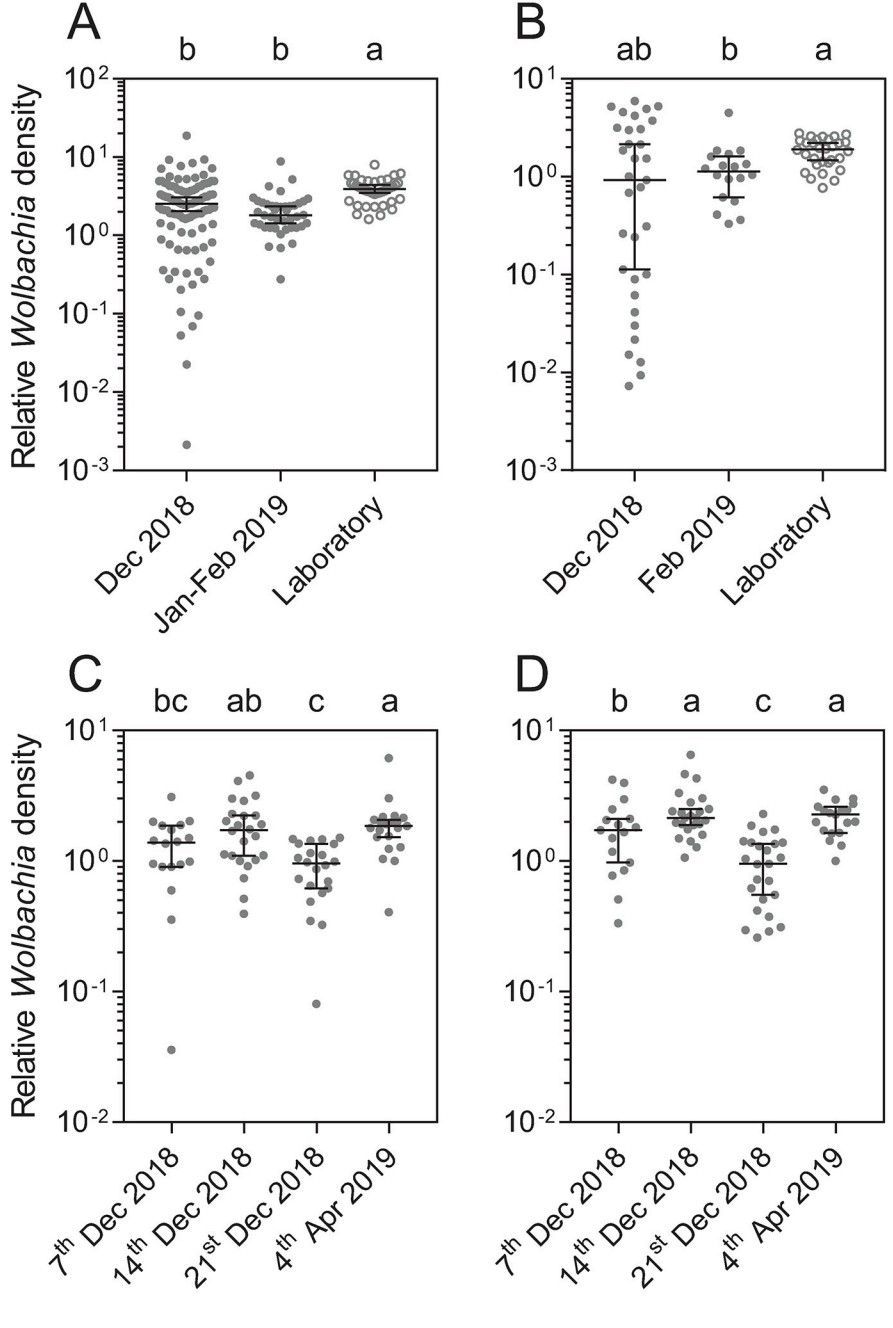

**Fig 3. *Wolbachia* density of field-collected or laboratory-reared *w*Mel-infected *Aedes aegypti*.** (A-B) *Wolbachia* density of field-collected (closed circles) and laboratory-reared (open circles) adults (A) and 4th instar larvae (B), where each dot represents the *Wolbachia* density of an individual mosquito (n = 18–105), averaged from three independent runs. Different letters indicate significant differences between collection dates by Dunn's multiple comparisons post-hoc tests. (C-D) *Wolbachia* density of (C) female and (D) male adults reared in the laboratory from field collected eggs. Each dot represents the median *Wolbachia* density of 2–6 individuals collected from a single ovitrap (n = 17–25). Different letters indicate significant differences between collection dates by Tukey's post-hoc test with Bonferroni correction. Bars are medians and 95% confidence intervals.

maternal transmission or the loss of *Wolbachia* from the previous generation. Our results indicate that the *w*Mel *Wolbachia* infection had been lost from a subset of the population following the heatwave, but later recovered.

*Wolbachia* densities in adults reared from ovitraps were variable, with substantial differences across collection dates (GLM: $F_{3,754}$ = 24.952, P < 0.001) and between ovitraps ($F_{102,754}$ = 2.612, P < 0.001) within each collection date. There was also an effect of sex, with males having higher *Wolbachia* densities overall than females ($F_{1,754}$ = 14.468, P < 0.001). This effect was due to lower Cp values for the *Ae. aegypti*-specific marker in females compared to males ($F_{1,921}$ = 55.031, P < 0.001) and not due to differences in Cp for the *w*Mel marker between the sexes ($F_{1,854}$ = 0.019, P = 0.891). *Wolbachia* densities were lowest in adults collected from the third week of sampling (Fig 3C and 3D), suggesting a delayed effect of the 26-29th of November heatwave on *Wolbachia* density in the subsequent generation.

## *Wolbachia* density is influenced by container location

To investigate variability in density effects across containers further, we measured *Wolbachia* densities from *Ae. aegypti* that were sampled from sentinel containers placed within the *w*Mel release zone in sunlight or full shade and colonized naturally. In this experiment, water temperatures in the bucket exposed to sunlight exceeded 41˚C while temperatures in full shade did not reach 30˚C during the monitoring period (Fig 4A). We found effects of container (GLM: $F_{1,75}$ = 102.064, P < 0.001) and date ($F_{3,75}$ = 30.310, P < 0.001) on *Wolbachia* density but there was no effect of sex ($F_{1,75}$ = 0.042, P = 0.838). Individuals sampled from the container in sunlight had *Wolbachia* densities that declined across collection dates (Fig 4B), with decreases occurring between the 1st and 4th of February when there were four consecutive days were water temperatures exceeded 40˚C. When mosquitoes were sampled at the same time from the containers in sunlight and full shade, *w*Mel densities were lower in mosquitoes sampled from the container exposed to sunlight (Fig 4B).

## Simulated heatwave effects on *Wolbachia* density are immediate with variable persistence into adulthood

We exposed *w*Mel-infected populations to simulated heatwaves across life stages and measured *Wolbachia* density immediately following exposure. *w*Mel density differed between heatwave treatments and across life stages (S1 Table). When reared in the absence of heat stress, *Wolbachia* density changed across life stages, with a high density in eggs that declined in early larvae and then increased over the course of larval and pupal development (Fig 5A). Exposure to heatwaves led to an immediate reduction in *Wolbachia* density in all life stages relative to unstressed controls, with a >96% reduction in density in early larvae, late larvae and pupae, and a 45.07% reduction in eggs (Fig 5A). When mosquitoes were exposed to heatwaves continuously over the course of their development, *Wolbachia* density declined as more stages were exposed (Fig 5A).

We measured the density of *w*Mel again when females and males exposed to simulated heatwaves during development had reached adulthood and reproduced. Females had higher densities than males (GLM: $F_{1,398}$ = 498.723, P < 0.001); we therefore analysed the sexes separately. There was an effect of exposure treatment on density in both females ($F_{4,180}$ = 70.508, P < 0.001) and males ($F_{4,218}$ = 250.625, P < 0.001). Heatwave exposures at all life stages reduced *Wolbachia* density in both female (Fig 5B) and male (Fig 5C) adults relative to unstressed controls. However, patterns of *Wolbachia* density in adults were inconsistent with those in immature stages that were tested immediately following exposure. *Wolbachia* densities in early and late instar larvae exposed to heatwaves were similar (Fig 5A), but in adults,

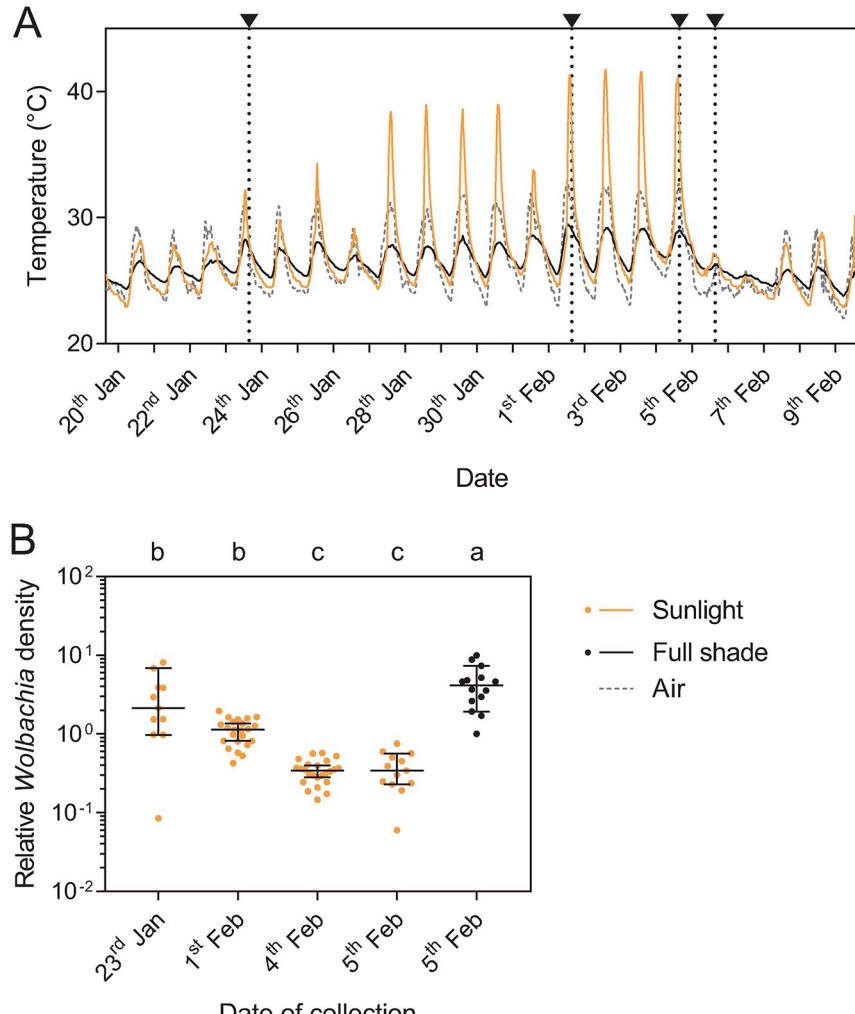

**Fig 4. Temperature cycles and *Wolbachia* densities in *Aedes aegypti* sampled from sentinel containers in the *w*Mel release zone.** (A) Water temperatures in buckets placed in an uncovered area exposed to sunlight (yellow) or in full shade (black) during January 2018 and measured at 30-minute intervals. Arrows and dotted vertical lines indicate time points when pupae were sampled. Air temperatures recorded at Cairns racecourse ([www.bom.gov.au](www.bom.gov.au)) are shown by the gray dotted line. (B) *Wolbachia* densities in adults collected as pupae from buckets that were in sunlight (yellow) or full shade (black). Different letters indicate significant differences between containers and collection dates by Tukey's post-hoc test with Bonferroni correction. Dots represent the *Wolbachia* density of an individual mosquito, averaged from three independent runs, while bars are medians and 95% confidence intervals.

densities were lower (by 91.83% in females and 95.85% in males) when early instar larvae were exposed to heatwaves compared to when late instar larvae were exposed (Fig 5B and 5C). This suggests that the ability of *w*Mel densities to recover may depend on when mosquitoes are exposed to heatwaves, with early instar larvae being the most vulnerable stage.

## Effects of simulated heatwaves on cytoplasmic incompatibility are life stage-specific

*w*Mel-infected males that were exposed to simulated heatwaves during different stages of development were crossed with uninfected females to see if effects on cytoplasmic incompatibility were life stage-specific. Unstressed *w*Mel-infected males induced complete cytoplasmic

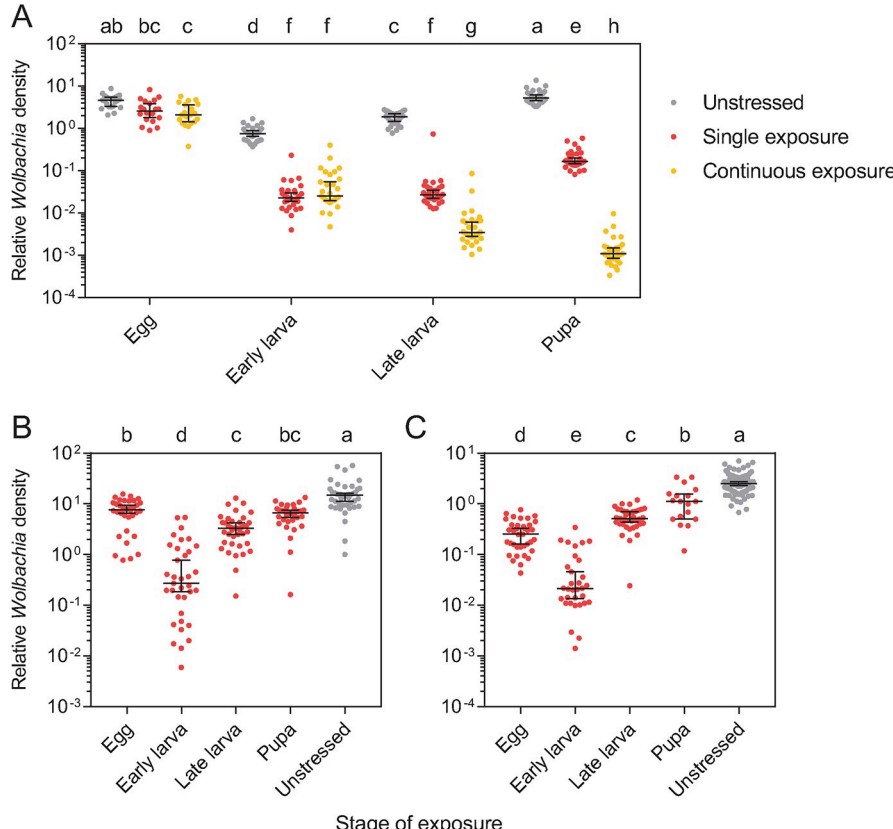

**Fig 5. *Wolbachia* density of *w*Mel-infected *Aedes aegypti* that were exposed to simulated heatwaves at different stages of development.** (A) *Wolbachia* density was determined for each life stage when mosquitoes were unstressed (gray), exposed to a single 72 hr heatwave event at each stage (red), or exposed to a heatwave continuously throughout development (yellow). (B-C) *Wolbachia* density was determined for (B) females and (C) males that were exposed to heatwaves at each life stage. Different letters indicate significant differences between treatments by Tukey's post-hoc test with Bonferroni correction. Dots represent the *Wolbachia* density of an individual mosquito, averaged from three independent runs, while bars are medians and 95% confidence intervals.

incompatibility with uninfected females, with zero eggs hatching (Fig 6A). Incomplete cytoplasmic incompatibility was observed when males were exposed to heatwaves at the egg, early larval, late larval and pupal stages (Fig 6A). Early instar larvae were the most severely affected, with similar hatch proportions (~0.5) to crosses where infected males were exposed to heatwaves for the entirety of their development (Mann-Whitney U: z = 0.144, P = 0.886). Pupae and late instar larvae were less susceptible, with *w*Mel-infected males exposed to heatwaves at these stages retaining stronger incompatibility (both z = 2.189, P = 0.029, Fig 6A).

We also tested the compatibility of *w*Mel-infected males with *w*Mel-infected females that were exposed to simulated heatwaves at each stage of development. Crosses between uninfected females and uninfected males where either males or females were exposed to heatwaves both exhibited high hatch proportions, indicating that this treatment does not affect female fertility (Fig 6B). Unstressed *w*Mel-infected females and males crossed to each other also had high hatch proportions, demonstrating that infected females can restore compatibility with infected males. *w*Mel-infected females that were stressed during the early larval instars had greatly reduced egg hatch compared to the unstressed control (96.34% reduction, Mann-Whitney U: z = 2.189, P = 0.029), indicating a loss of ability to restore compatibility with infected males. Other stages were less severely affected, with moderate reductions in egg hatch when

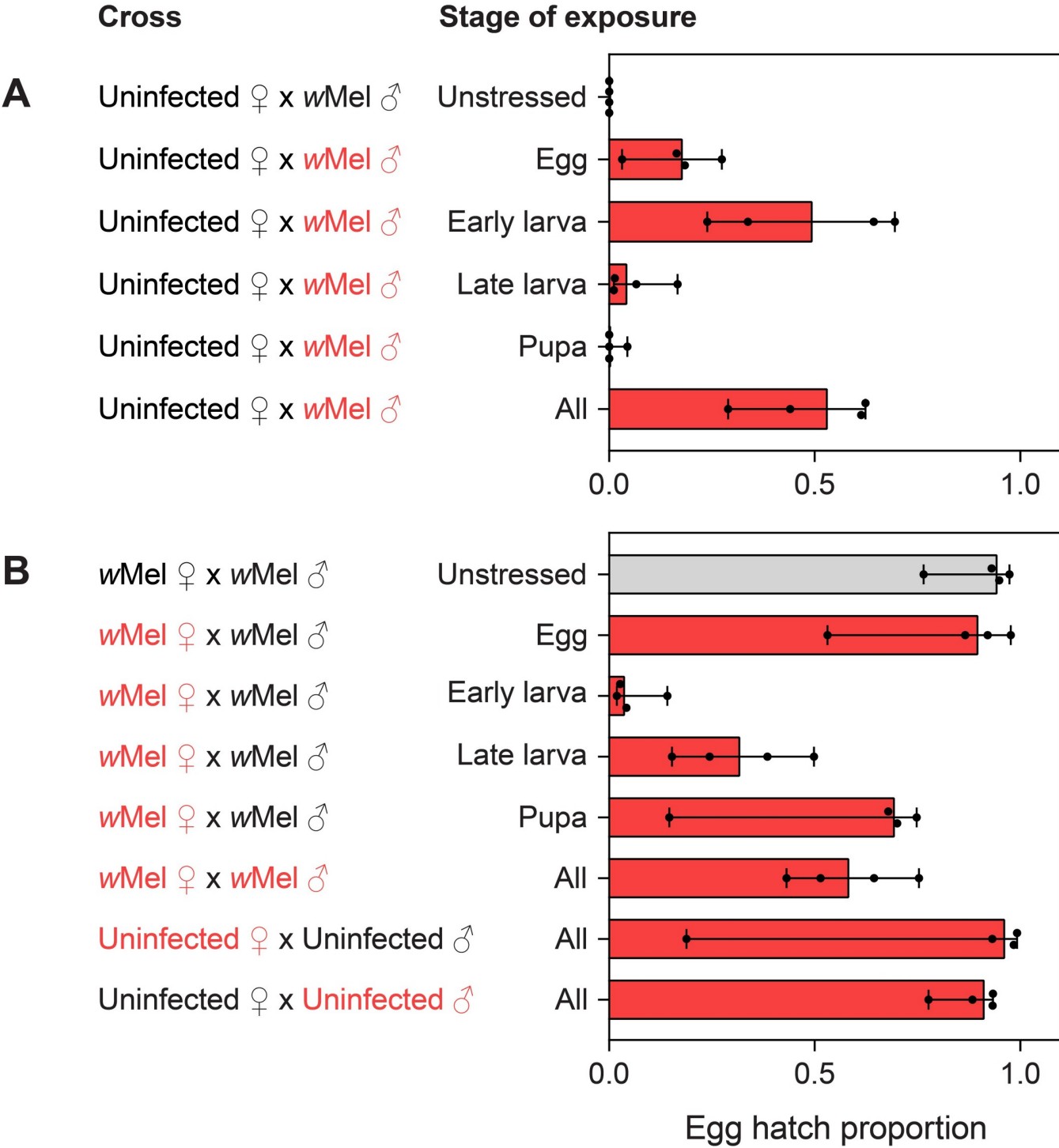

**Fig 6. Egg hatch proportions from crosses with *w*Mel-infected *Aedes aegypti* that were exposed to heatwaves at different stages of development.** Crosses are separated by theoretically incompatible crosses (A) and theoretically compatible crosses (B). Red text denotes the sexes that were exposed to heatwaves in each cross. Bars are medians and error bars are 95% confidence intervals, with dots showing average egg hatch proportions from a single replicate cage (n = 4 cages per treatment and 10 adults of each sex per cage).

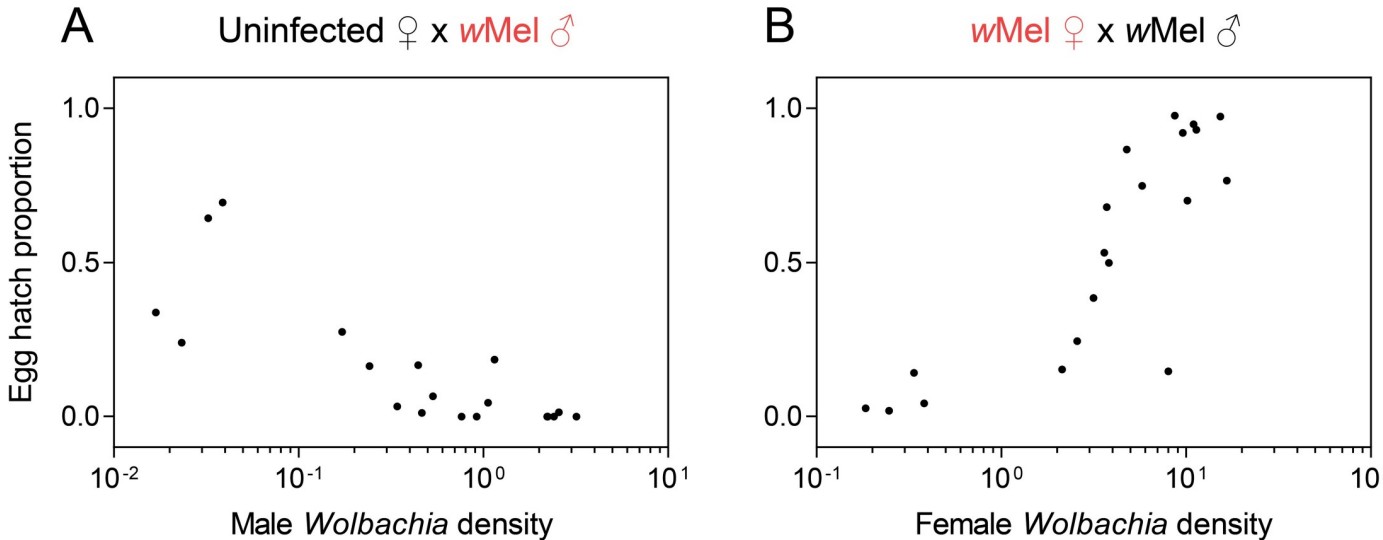

**Fig 7. Relationships between *Wolbachia* density and the fidelity of cytoplasmic incompatibility.** (A) The relationship between male *Wolbachia* density and cytoplasmic incompatibility induction when *w*Mel-infected males were exposed to various heatwave treatments. (B) The relationship between female *Wolbachia* density and the ability of females to restore compatibility with infected males. Each data point represents the egg hatch proportion from a single replicate cage and the average *Wolbachia* density from all individuals tested in the cage. Red text denotes the sex that was exposed to different temperature treatments in each cross.

*w*Mel-infected females were exposed to heatwaves during the late larval (66.51% reduction) and pupal (26.59% reduction) stages (both z = 2.189, P = 0.029) and no significant effect when eggs were exposed (z = 0.405, P = 0.686). When both sexes were *w*Mel-infected and exposed to heatwaves throughout development, hatch proportions were intermediate, resulting from loss of male ability to induce cytoplasmic incompatibility, loss of female ability to restore compatibility, or a combination of both.

## Cytoplasmic incompatibility is *Wolbachia* density-dependent

We determined average *Wolbachia* densities for each replicate cage in the experimental crosses and were therefore able to examine relationships between *Wolbachia* density and egg hatch proportions. The ability of males to induce cytoplasmic incompatibility decreased as *Wolbachia* density decreased, with a strong negative association between male *Wolbachia* density and egg hatch proportion in this cross (Spearman's rank-order correlation: ρ = -0.801, P < 0.001, n = 20, Fig 7A). Similarly, the ability of females to restore compatibility with males that had a high density (Fig 5C) decreased as female *Wolbachia* density decreased, with a strong positive association between female density and egg hatch proportion (ρ = 0.859, P < 0.001, n = 20, Fig 7B). These results demonstrate a direct relationship between *Wolbachia* density and cytoplasmic incompatibility and rescue phenotypes.

## Discussion

Here we show that a heatwave in Cairns, Australia, led to a reduction in *w*Mel frequencies and densities in an *Ae. aegypti* population. Prior studies have demonstrated the loss of *w*Mel under cyclical laboratory conditions that include hot periods [7, 17, 22] and the current study shows that this can also occur in natural populations. However, the heatwave reduced or eliminated *Wolbachia* from only a subset of mosquitoes, and both the frequency of *Wolbachia* and density effects recovered relatively quickly after the heatwave. The infection frequency remained well above the threshold required for invasion, estimated to be ~20% for *w*Mel [11], which should

result in *Wolbachia* infection frequencies increasing rapidly through cytoplasmic incompatibility [5] after a heatwave has passed. Thus, heatwaves of the magnitude experienced in Cairns are unlikely to have long-term effects on *w*Mel release programs once the infection has established in the population, although it may make invasion more difficult in the first instance and may inform timing of releases in areas where heatwaves are common. Our results also suggest that *w*Mel may be resilient in the face of heatwaves caused by climate change [24]. However, ongoing monitoring following releases [8, 13, 34] may be necessary in areas where hot periods are common, and/or where common larval habitats are exposed to sunlight and larvae are routinely exposed to high temperatures.

The different infection frequencies and densities between life stages in our field monitoring suggest that heatwave effects are life stage-specific. Ulrich et al. [18] previously demonstrated that effects on adult *w*Mel density depend on the life stage exposed, with cyclically high temperatures that coincided with the 1st and 2nd larval instars tending to have greater effects. We show that exposure of early instar larvae to simulated heatwaves also has the strongest effects on cytoplasmic incompatibility, both in terms of male ability to induce cytoplasmic incompatibility and female ability to restore compatibility with infected males. In the field, young larvae may therefore be most vulnerable to losing the *w*Mel infection. Reduced infection frequencies in field-collected larvae but not adults following the heatwave provides some support for this hypothesis. However, we did not have information on the life stage of the sampled adults and larvae during the heat wave, because we were only able to sample *Ae. aegypti* after the heat wave, and development of mosquito larvae is sharply affected by larval density and food availability [35, 36]. *Wolbachia* loss can occur either within [22] or across [23] generations, with potential effects on *Wolbachia* frequency and density several weeks after a heatwave. *w*Mel density can also vary throughout adulthood, tending to increase with age under laboratory conditions [7, 18] and can vary depending on the history of the previous generations [20]. Any effects of heatwaves on density may therefore be indirect as well as direct, such as through changing the age distribution of adults or conditions experienced in the previous generation.

The laboratory experiments also highlight the importance of adult *Wolbachia* densities in affecting phenotypes associated with *Wolbachia*, with lower densities corresponding to weaker cytoplasmic incompatibility induction by males and reduced ability of females to restore compatibility with infected males. The two phenotypes appear to require different whole-adult *w*Mel densities; at a relative *Wolbachia* density of 1, males induced near-complete cytoplasmic incompatibility while females had largely lost their ability to restore compatibility with infected males. This pattern is consistent with a previous study in *Ae. aegypti* [22] and *Drosophila* research showing that males from infected lines can induce incompatibility even when the males are very weakly infected, or lack detectable levels of *Wolbachia* [37]. Therefore, under certain conditions in the field, *w*Mel-infected populations may become partially self-incompatible. Given that cytoplasmic incompatibility is strong at relatively low densities and that *w*Mel appears to be more resilient to heat in adults, heatwaves are unlikely to affect the success of *Wolbachia*-based population suppression programs that rely on the release of male adults to induce cytoplasmic incompatibility.

The laboratory experiments highlight that effects of sporadic high temperatures on *w*Mel densities at one life stage do not necessarily translate into later stages. Exposure of eggs to the heatwave treatment had a modest immediate effect on density, but effects persisted into adulthood with males inducing incomplete cytoplasmic incompatibility. In contrast, when early and late instar larvae were exposed to heatwaves, there was a substantial and immediate decrease in density in both cases, but *w*Mel densities recovered substantially in the late instar treatment. These results suggest that heatwave effects on *Wolbachia* density interact with mosquito development in subtle ways.

The present results focus on the *w*Mel infection, whereas heatwave effects will undoubtedly depend on the nature of the *Wolbachia* strains and their hosts. Over 25 *Wolbachia* infection types have been generated in pathogen-transmitting mosquitoes through microinjection and these exhibit variable effects on mosquito fitness, pathogen blocking and cytoplasmic incompatibility [1]. Only two of these strains, *w*AlbB and *w*Mel, have so far been released and established in natural populations in replacement programs [9, 11], so there are few opportunities to compare the performance of strains in the field. Direct comparisons under natural conditions are important given that environmental conditions can have differential effects on *Wolbachia* strains [17, 38]. *Wolbachia* density variation resulting from heatwaves may have important implications for pathogen blocking by *Wolbachia* [39, 40], but this remains to be tested within a field context.

## Supporting information

**S1 Table. General linear model for immediate effects of simulated heatwaves on *w*Mel density across life stages in *Aedes aegypti*.**
(DOCX)

## Acknowledgments

We thank Michael Townsend, Jason Anderson, Gerhard Ehlers, Rod Bellwood and Barry Bennett for assistance with field sampling.

## Author Contributions

**Conceptualization:** Perran A. Ross, Jason K. Axford, Scott A. Ritchie, Ary A. Hoffmann.

**Data curation:** Perran A. Ross.

**Formal analysis:** Perran A. Ross.

**Funding acquisition:** Scott A. Ritchie, Ary A. Hoffmann.

**Investigation:** Perran A. Ross, Jason K. Axford, Qiong Yang, Kyran M. Staunton, Scott A. Ritchie, Kelly M. Richardson.

**Methodology:** Perran A. Ross, Jason K. Axford, Qiong Yang, Kyran M. Staunton, Scott A. Ritchie, Ary A. Hoffmann.

**Resources:** Kyran M. Staunton, Scott A. Ritchie.

**Visualization:** Perran A. Ross.

**Writing – original draft:** Perran A. Ross.

**Writing – review & editing:** Perran A. Ross, Jason K. Axford, Qiong Yang, Kyran M. Staunton, Scott A. Ritchie, Kelly M. Richardson, Ary A. Hoffmann.

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
