## [Decision Letter · Decision Letter 0]

12 Nov 2019

Dear Dr. Ross:

Thank you very much for submitting your manuscript "Heatwaves cause fluctuations in wMel Wolbachia densities and frequencies in Aedes aegypti" (PNTD-D-19-01580) for review by PLOS Neglected Tropical Diseases. Your manuscript was fully evaluated at the editorial level and by independent peer reviewers. The reviewers appreciated the attention to an important topic but identified some aspects of the manuscript that should be improved.

We therefore ask you to modify the manuscript according to the review recommendations before we can consider your manuscript for acceptance. Your revisions should address the specific points made by each reviewer.

(1) A letter containing a detailed list of your responses to the review comments and a description of the changes you have made in the manuscript.

(2) Two versions of the manuscript: one with either highlights or tracked changes denoting where the text has been changed (uploaded as a "Revised Article with Changes Highlighted" file ); the other a clean version (uploaded as the article file).

(3) If available, a striking still image (a new image if one is available or an existing one from within your manuscript). If your manuscript is accepted for publication, this image may be featured on our website. Images should ideally be high resolution, eye-catching, single panel images; where one is available, please use 'add file' at the time of resubmission and select 'striking image' as the file type. 

Please provide a short caption, including credits, uploaded as a separate "Other" file. If your image is from someone other than yourself, please ensure that the artist has read and agreed to the terms and conditions of the Creative Commons Attribution License at http://journals.plos.org/plosntds/s/content-license (NOTE: we cannot publish copyrighted images). 

(4) Appropriate Figure Files 

Please remove all name and figure # text from your figure files upon submitting your revision. Please also take this time to check that your figures are of high resolution, which will improve both the editorial review process and help expedite your manuscript's publication should it be accepted. Please note that figures must have been originally created at 300dpi or higher. Do not manually increase the resolution of your files. For instructions on how to properly obtain high quality images, please review our Figure Guidelines, with examples at: http://journals.plos.org/plosntds/s/figures

While revising your submission, please upload your figure files to the Preflight Analysis and Conversion Engine (PACE) digital diagnostic tool, https://pacev2.apexcovantage.com/ PACE helps ensure that figures meet PLOS requirements. To use PACE, you must first register as a user. Then, login and navigate to the UPLOAD tab, where you will find detailed instructions on how to use the tool. If you encounter any issues or have any questions when using PACE, please email us at figures@plos.org.

We hope to receive your revised manuscript by Jan 11 2020 11:59PM. If you anticipate any delay in its return, we ask that you let us know the expected resubmission date by replying to this email.

To submit your revised files, please log in to https://www.editorialmanager.com/pntd/

Sincerely,

Alain Kohl

Guest Editor

Scott Weaver

Deputy Editor

A number of minor suggestions that the authors should address, were made by the reviewers.

Reviewer's Responses to Questions

**Key Review Criteria Required for Acceptance?**

**Methods**

-Are the objectives of the study clearly articulated with a clear testable hypothesis stated?

-Is the study design appropriate to address the stated objectives?

-Is the population clearly described and appropriate for the hypothesis being tested?

-Is the sample size sufficient to ensure adequate power to address the hypothesis being tested?

-Were correct statistical analysis used to support conclusions?

-Are there concerns about ethical or regulatory requirements being met?

Reviewer #1: The objectives are clear, with testable hypotheses. 

The studies have been robustly carried out with good statistical treatment. 

There are no ethical concerns.

Reviewer #2: The methods are well described with clear objectives. The authors took advantage of the heat period faced in North Queensland and used their previous knowledge about the effects of high temperatures on the interaction between Aedes aegypti and Wolbachia strains to design experiments to further address such relations using field data.

Reviewer #3: Can more information be given about the trapping methods? Particularly the prototype acoustic traps. 

L258 also refers to BGS traps, but this is not mentioned previously.

**Results**

-Does the analysis presented match the analysis plan?

-Are the results clearly and completely presented?

-Are the figures (Tables, Images) of sufficient quality for clarity?

Reviewer #1: Tables and figures are clear and well described in written component.

Reviewer #2: The results are well presented, with adequate statistical analysis and Figures/tables presenting the information with clarity.

Reviewer #3: L287. Was the higher density in males due to an increase in the CT of the Wolbachia PCR, or a decrease in the host gene. Given males are smaller this could explain the higher ratio of Wolbachia to host.

Can the heat wave related days in figure 1 be shaded or indicated by some means?

Figure 3. I take it there is no significant difference in the treatments within A and B? Could this be indicated?

**Conclusions**

-Are the conclusions supported by the data presented?

-Are the limitations of analysis clearly described?

-Do the authors discuss how these data can be helpful to advance our understanding of the topic under study?

-Is public health relevance addressed?

Reviewer #1: Conclusions are well supported. 

Good discussion of the potential impact of the findings for release programs more generally.

Reviewer #2: Conclusions are supported by the data presented. Most importantly, the authors seek to simulate natural variation on temperature acting on male and female Aedes aegypti in regards to Wolbachia attributes such as cytoplasmic incompatibility and Wolbachia density. The observed effects might also affect invasion pattern of Wolbachia, since it has now being released in at least 14 countries worldwide.

Reviewer #3: The discussion is rather short, and does not mention how reductions in density by temperature could affect pathogen transmission. There are studies in Anopheles system that examine the role of temperature and pathogen transmission that could be discussed.

Furthermore, there appear to be quite a large variation in the density of some samples (Dec 2018). This could influence vector competence. This could be discussed in further detail bringing in the work of Gabriela Gomes which examines heterogenous effects of infections.

**Editorial and Data Presentation Modifications?**

Reviewer #1: The paper is very well written.

Reviewer #2: Authors present robust evidence about the role of temperature (specially heatwaves) on the cytoplasmic incompatibility phenotype and Wolbachia density of Aedes aegypti. I have only a couple of suggestions of minor modifications before publication:

1. Abstract and Author Summary. Although the experiments were done in North Queensland, using mosquitoes with an Australian background, I would recommend one sentence to address the fact Wolbachia deployment has been ongoing in several countries and each one is likely to face heatwaves as well. So, results observed using Australian landscape (local mosquitoes, local temperatures, local effects), those outcomes on cytoplasmic incompatibility and Wolbachia density might be seen elsewhere, showing the effects of high temperatures on maintaining long-term attributes of Wolbachia is a worth theme to investigate further.

2. Line 108. The last sentence of Introduction section seems too vague. Precisely how those results on life stage-specific temperature effects on cytoplasmic incompatibility and Wolbachia density would inform Wolbachia release programs. Such affirmation requires additional clarity.

3. Lines 118 and 127: The peak of temperature on Cairns was observed on late November and adult Aedes aegypti collection started on December 2018. It would add clarity if the precise dates of adult collection are provided by authors in order to see the exact temperature during field collections.

4. Line 131: Considering only 4th instar larvae and adults were used for Wolbachia density, larvae from other instar and pupae were discarded?

5. Line 141: What was the number of eggs collected with ovitraps? Information such as total number or average number would be important. Furthermore, what is the mean distance among ovitraps? How representative of local population the 40 ovitraps might be?

6. Line 155: The description of trapping with 150 ovitraps from February to March 2018 to produce baseline information should come before the later effort to check Wolbachia density.

7. Line 282: The fact one single ovitrap had all tested individuals negative to Wolbachia, in an area in which Wolbachia frequency is close to 100% is unexpected. Negative controls of PCR reaction worked fine? What was the distance between this negative ovitrap and the other closest one? Any local particularity worth mentioning as insecticide use by this householder, presence of pets, etc?

8. Line 287 and 308: The relation between temperature and Wolbachia density seems to have different outcomes regarding mosquito sex if we go for adults reared from ovitraps, but was effectless if looking for Ae. aegypti sampled in sentinel containers. Finally, in lab assays in which the stage exposed to heatwaves varied, females had higher densities than males. Therefore, have all data in hands, each one pointing to a different relation between Wolbachia density and mosquito sex, what would be the general conclusion regarding this interaction?

9. Line 473: Authors highlighted heatwaves effects on the long-term effectiveness of Wolbachia to mitigate disease transmission, especially under replacement program strategies. However, what would be expected effects on suppression strategies? I believe it is worthwhile to add a few sentences to consider the use of Wolbachia deployment for suppression.

10. Figure 1: Consider highlighting (or maybe shading) the period in which adult collections were performed on the field to promote a straight comparison of field temperatures and the time in which collections were done. 

11. Figure 2: I might have lost it, but why not replicate the same temperature stress in Wolbachia-uninfected individuals?

12. Figure 3. More information would be available if authors added the number of mosquitoes tested, preferentially in the legend to not pollute the graphs.

Reviewer #3: Should links be used in the main text?

**Summary and General Comments**

Reviewer #1: This is an interesting study examining the impact of a heat wave in Cairns, Australia on wMel infection frequencies in Aedes aegypti post field release. Previous manipulative lab or pseudo field experiments have identified that heightened temperature may lead to reduced maternal transmission. This study uniquely captures this effect in a natural setting. The authors demonstrate ~12% reduction in Wolbachia infection frequencies that took approximately 4 months to recover. They also show that the time of the heat exposure during development, eggs, different larval instars, etc have varying impacts on the Wolbachia loads in the adult, where CI and pathogen blocking expression are relevant. The data have consequences for the handful of scientists/decision makers running release programs globally. As more strains are tested there may be motivation to deploy something other than wMel in regions with consistently higher temperatures. 

Questions the authors may want to consider addressing:

The paper nicely includes a manipulative component demonstrating the impact of similar temperatures. It is a shame that baseline monitoring was not done in the same season up until the time just before the heat wave. That dampens the evidence for cause and effect a bit but as a whole package the evidence from the manuscript is strong. 

Is there any evidence from any previously published post release monitoring of lower Wolbachia loads in summer? I am curious whether there is a threshold effect of temperature and duration that has to occur before there is an impact or whether there is a more linear relationship with rising temperatures? Your bucket temp experiment suggests maybe 40 degrees at least is a threshold. 

What do you think explains the different responses to heat at the different developmental stages? Interactions with immunity or rate of host cell division, etc?

Reviewer #2: The manuscript entitled "Heatwaves cause fluctuations in wMel Wolbachia densities and frequencies in Aedes

aegypti", by Ross et al., presents significant data regarding the effects of temperature on Wolbachia densities in Aedes aegypti mosquitoes from North Queensland. Considering Wolbachia deployment has been currently conducted in at least 14 countries, each one being likely to experience heatwaves, results have a timely relevance that support its publication after some small questions are addressed. Overall, results were obtained after week-described and designed experiments and the conclusions are supported by the data gathered in such assays.

Reviewer #3: The work by Ross et al examined the effect of heat waves on Wolbachia infections in Aedes aegypti. Given the changing nature of the climate, studies of this nature are critical to understand the success of interventions such as the deployment of Wolbachia for control of arboviruses. As such the paper will be of general interest to the readers of PLOS NTD. In general, the work is thorough and the comments here should be addressable before publication. 

L471. Pathogen-transmitting. Not disease-vectoring. Disease is the symptom(s) that occur in the susceptible host, not the mosquito.

Fig 4. Examines breeding sight in sunlight or shade. What are the preferences for females to lay in these sights? One might expect females to lay in sight more suitable for development of their larvae. I presume females laid into these traps before the heat wave?

Can there be some discussion as to the single ovitrap that found 100% Wolbachia negative samples. Was this in the region where Wolbachia would be expected to be found? 

Fig 6. Can the sample size per cage be included in the figure legend?

PLOS authors have the option to publish the peer review history of their article (what does this mean?). If published, this will include your full peer review and any attached files.

Reviewer #1: No

Reviewer #2: Yes: Rafael Maciel-de-Freitas

Reviewer #3: No

---

## [Editor Report · Decision Letter 1]

27 Nov 2019

Dear Dr. Ross,

We are pleased to inform you that your manuscript, "Heatwaves cause fluctuations in wMel Wolbachia densities and frequencies in Aedes aegypti", has been editorially accepted for publication at PLOS Neglected Tropical Diseases.

Before your manuscript can be formally accepted and sent to production you will need to complete our formatting changes, which you will receive in a follow up email. Please note: your manuscript will not be scheduled for publication until you have made the required changes.

IMPORTANT NOTES

* Copyediting and Author Proofs: To ensure prompt publication, your manuscript will NOT be subject to detailed copyediting and you will NOT receive a typeset proof for review. The corresponding author will have one final opportunity to correct any errors when sent the requests mentioned above. Please review this version of your manuscript for any errors.

* If you or your institution will be preparing press materials for this manuscript, please inform our press team in advance at plosntds@plos.org. If you need to know your paper's publication date for media purposes, you must coordinate with our press team, and your manuscript will remain under a strict press embargo until the publication date and time. PLOS NTDs may choose to issue a press release for your article. If there is anything that the journal should know, please get in touch.

*Now that your manuscript has been provisionally accepted, please log into EM and update your profile. Go to http://www.editorialmanager.com/pntd, log in, and click on the "Update My Information" link at the top of the page. Please update your user information to ensure an efficient production and billing process.

*Note to LaTeX users only - Our staff will ask you to upload a TEX file in addition to the PDF before the paper can be sent to typesetting, so please carefully review our Latex Guidelines [http://www.plosntds.org/static/latexGuidelines.action] in the meantime.

Best regards,

Alain Kohl

Guest Editor

Scott Weaver

Deputy Editor

---

## [Editor Report · Acceptance letter]

7 Jan 2020

Dear Dr. Ross,

We are delighted to inform you that your manuscript, "Heatwaves cause fluctuations in wMel *Wolbachia* densities and frequencies in *Aedes aegypti*," has been formally accepted for publication in PLOS Neglected Tropical Diseases.

Best regards,

Serap Aksoy

Editor-in-Chief

Shaden Kamhawi

Editor-in-Chief
